# Esophago-Gastric Tube (EG Tube) in Debilitated Sea Turtle Rehabilitation: Insight in 195 Loggerheads *Caretta caretta*, Southern Italy

**DOI:** 10.3390/vetsci11010035

**Published:** 2024-01-15

**Authors:** Antonio Di Bello, Delia Franchini, Stefano Ciccarelli, Daniela Freggi, Francesco Caprio, Pasquale Salvemini, Carmela Valastro

**Affiliations:** 1Department of Veterinary Medicine, University of Bari, SP 62 per Casamassima Km 3, 70010 Valenzano, BA, Italy; antonio.dibello@uniba.it (A.D.B.); delia.franchini@uniba.it (D.F.); daniela.freggi@uniba.it (D.F.); francesco.caprio@uniba.it (F.C.); carmela.valastro@uniba.it (C.V.); 2Lampedusa Sea Turtle Rescue Center, 92031 Lampedusa, AG, Italy; 3WWF Molfetta Rescue Center, Via Puccini 16, 70056 Molfetta, BA, Italy; crtmmolfetta@gmail.com

**Keywords:** sea turtle, esophago-gastric tube, nutritional support, rehabilitation

## Abstract

**Simple Summary:**

This study examined the use of a permanent esophago-gastric tube (EG tube) in sea turtles as an alternative to force-feeding. The study included 195 marine turtles that had surgery to place a permanent EG tube between 2008 and 2022. Loggerhead turtles (*Caretta caretta*) needed EG tube placement due to their poor health for various diseases, which were categorized. The study analyzed the duration of EG tube placement in relation to the animals’ condition and injuries, considering any complications or differences between the two facilities.

**Abstract:**

Efficient nutritional support plays a pivotal role in the rehabilitation of sea turtles, ensuring a positive, swift, and successful recovery from clinical conditions for their reintroduction into the wild. For sea turtles in severely emaciated and underweight condition, the primary objective is to facilitate weight gain in terms of both muscle mass and fat reserves. Traditionally, many sea turtle rehabilitators have employed the practice of force-feeding, which also involves the daily insertion of an orogastric tube from the mouth to the stomach. However, this technique may be highly stressful for the animals, carrying the risks of regurgitation and potential harm, requiring the removal of the animal from the water, and subjecting it to uncomfortable and potentially dangerous handling. The procedure may also involve risks for operators. In this study, we explore the utilization of a permanent esophago-gastric tube (EG tube) in sea turtles as an alternative to force-feeding, providing a respectful and appropriate approach to meeting the nutritional needs of patients. The administration of food, essential medications, and fluids is performed directly with the turtle in its tank, minimizing the stress associated with handling, while ensuring the safety of operators. The study involves 195 marine turtles that underwent surgery for the placement of a permanent EG tube between 2008 and 2022. Of these, 116 animals were treated at the Sea Turtle Clinic of the Department of Veterinary Medicine at the University of Bari, South Adriatic Sea, Puglia (IT), and 79 patients were admitted to the Sea Turtle Rescue Center of Lampedusa, Central Mediterranean Sea, Sicily (IT). The loggerhead turtles (*Caretta caretta*) required EG tube placement due to their poor condition related to various diseases, which were systematically categorized. The duration of EG tube placement was analyzed regarding the specific condition of the animals and the nature of their injuries, considering any complications or differences between the two facilities. The results of the study will provide valuable information for the ongoing care and treatment of marine turtles in rehabilitation facilities.

## 1. Introduction

Addressing the needs of underweight sea turtles involves a multidisciplinary approach, including collaboration between veterinarians, conservationists, and rehabilitation experts. Regular monitoring and research are essential to better understand and mitigate the factors leading to compromised conditions in sea turtles [1]. Their conditions can be attributed to various factors: interactions with fishing equipment resulting in severe injuries across their bodies, pollution-related issues such as entanglement, collisions causing extensive injuries, or pulmonary diseases [2,3,4]. The animals can present a debilitated condition also because of complex surgical procedures, experiencing pain, or stress associated with captivity. Generally, emaciated turtles typically do not eat voluntarily and there is no way to induce them to eat. An appropriate and energetically balanced diet consisting of proteins, lipids, carbohydrates, fibers, minerals, vitamins, fatty acids, and amino acids [5] plays a vital role in any rehabilitation effort. The absorption of nutrients from food is essential for the proper functioning of many physiological processes and drug metabolism, which is significant for the improvement of the health of the patient. In the case of sea turtles in very thin and underweight condition, the primary objective is to promote weight gain in terms of muscle mass and fat reserves [6]: efficient nutritional support to sea turtles under rehabilitation is an important step to ensure a positive and faster recovery from clinical conditions, to reintroduce the animals in the wild. Previous studies have indicated that loggerheads are opportunistic carnivores, primarily consuming benthic invertebrates and fish [7]. Many sea turtle rehabilitators have historically used and still use the practice of force-feeding, inserting daily an orogastric pipe from the mouth to the stomach to administer a fish-based smoothie [8,9]. This technique is highly stressful for the animals, with possible regurgitation, potential damage to the esophagus and/or the digestive tract, and in worst cases the rhamphotheca [10]. It is not applicable to animals that have had surgery to their esophagus. Stress and risks are repeated every time the turtle is fed, requiring moving the animal out of the water and handling it in a very uncomfortable position. Furthermore, this practice can pose significant danger to the operators, who risk being bitten or injured by the distressed patient.

Existing literature has previously discussed how debilitated reptiles often may require a long period of supportive care and treatment [11,12]: the placement of an esophago-gastric tube is commonly performed in chelonian tortoises, for offering nutritional support during periods of anorexia or facilitating the oral administration of medications. Regrettably, very little information is available related to the frequency and types of complications observed in these patients [13]. Studies conducted in dogs and cats have revealed that naso-gastric gastrotomy tubes placed endoscopically [14] present complication rates of 37% and 62.5%, respectively. 

In this study, the placement of a permanent esophago-gastric tube in sea turtles is described as a valid substitute for force-feeding and a respectful and appropriate procedure for marine turtles, which greatly simplifies the daily feeding process by allowing food to be administered with the animal directly into the water of its tank, minimizing the stress associated with handling. Additionally, essential drugs (antibiotics, vitamins, etc.) and fluids which are crucial for their clinical management are easily administered through this route [15]. Additionally, it is essential to highlight that this method is entirely safe and poses no danger to operators either.

We conducted a comprehensive analysis of data spanning 15 years (2008–2022) concerning 195 marine turtles that underwent surgery to have a permanent esophago-gastric tube (EG tube) implanted to support their nutritional needs: 116 animals were treated at the Sea Turtle Clinic of the Department of Veterinary Medicine of the University of Bari, situated along the South Adriatic Sea in Puglia (IT), and an additional 79 patients were admitted to the Sea Turtle Rescue Center of Lampedusa, located in the Central Mediterranean Sea off the coasts of Sicily (IT). Over the years, the surgical technique [15] has been continually refined. The loggerhead turtles (*Caretta caretta*) requiring EG tube placement were affected by various diseases which were systematically categorized. We collected data to analyze the duration of the EG tube usage in relation to their specific condition, and the type of injuries they suffered and to discern any discrepancies between the two facilities. 

## 2. Materials and Methods

From January 2008 to June 2022, a total of 195 loggerhead sea turtles (*Caretta caretta*) received medical attention. Specifically, 116 loggerheads provided by various rescue centers across South Italy were treated at the Sea Turtle Clinic (STC) of the Department of Veterinary Medicine of University of Bari, Puglia bordering the Adriatic Sea; the remain 79 turtles were rescued and treated at the Lampedusa Sea Turtle Rescue Center (LSTR), Sicily, situated in the heart of the Central Mediterranean Sea.

The sea turtles had been accidentally caught with trawls or gill nets, hooked with longlines, caught with floating driftnet, stranded, or entangled. 

Due to incomplete data for 44 turtles, they were excluded from the analyses. Among the remaining 151 loggerheads, 91 individuals exhibited the presence of hooks and/or fishing lines in various parts of their gastrointestinal tracts, while 26 turtles had suffered from skull or carapace fracture. Additionally, 34 turtles displayed a particularly deteriorated condition, in 14 cases related to entanglement of their flippers or neck (Figure 1).

Each patient underwent an assessment of their body condition score (BCS), which provides a subjective evaluation of their physical condition [16], employing a scale ranging from 1 (emaciated) to 5 (obese). During their evaluation, any observed complications were documented. Morphometric measurements were recorded: the curved carapace length from notch to tip (CCL), the curved carapace width (CCW), and the weight. 

All of the turtles underwent radiographic examinations in the three projections: dorsoventral (DV) vertical beam, latero-lateral (LL), and craniocaudal (Cr-Cd) horizontal beam [17,18]. Ultrasound examinations were conducted for those animals displaying lines from the mouth or cloaca, and for turtles in which radiographic exams revealed the presence of one or more hooks lodged in various segments of the digestive tract. 

Blood samples were collected from each turtle to measure plasma biochemical and hematological values in order to complete the general health status assessment; unfortunately, LSTR occasionally encountered difficulties in consistently executing this procedure.

The diagnostic findings were instrumental in determining the suitability of the adjunctive procedure to place the EG tube, which was assessed during the scheduled surgery. This evaluation considered not only the significance of the BCS but also the overall clinical conditions, taking in account the presence of skull lesions as well as the presence of hooks and lines in the gastro-intestinal tract. The duration of the EG tube placement for each animal was documented and analyzed separately for animals that did not survive and patients that successfully recovered, followed by a global assessment. 

Turtles for which complete information was not available were excluded from this study. 

### 2.1. Surgical Technique

In many cases, the placement of the EG tube was performed before the completion of the necessary surgical procedures required by each animal (extraction of lines/hooks, amputation…), generally occurring on the day of the patient’s arrival or the following day; in several other cases, a dedicated surgical session was scheduled in order to place the EG tube, considering the critical condition of the patient. 

Levin’s gastric catheters were used, with a diameter from 12 Fr to 24 Fr, depending to the size of the animal, and with a length of 75 or 120 cm. In large turtles weighing more than 40–45 kg, transparent non-toxic food grade PVC pipes [10,15] were employed, available with a diameter of 10 to 14 mm, cut to the required length. Before the surgical procedure, the portion of the EG tube to be inserted was carefully marked, using the measurement of the distance between the medial third of the cervical region and the left lateral margin of the carapace, taken at the widest point on the turtle while in the ventral position. This measurement allowed us to project the location of the gastric area onto the tube accurately. In the situations requiring any prior surgical procedure, the EG tube was inserted upon completion of the ongoing operation. The selection of anesthetic protocols was determined based on the type of surgery, the patient’s condition, and the expected duration of the procedure. The induction phase involved the administration of 5 mg/kg IV of propofol (PropoVet 10 mg/mL, Zoetis S.r.l. Italia), preceded where feasible, based on the animal’s condition, by 5 mg/kg IV of tramadol (Altadol 50 mg/mL, FormeVet S.r.l., Italia). Anaesthesia was consistently maintained using oxygen and 2–3% isoflurane (IsoFlo, Zoetis SrL, Italia) [19].

In the case of severely debilitated sea turtles, which included skull and carapace fractures, needing surgery solely for the placement of the EG tube, anesthesia was induced with a bolus of 5–7 mg/kg IV propofol.

The insertion of the EG tube may be considered a quite rapid procedure: following endo-tracheal intubation, the turtle was placed in dorsal recumbency, with the neck fully extended and the head positioned slightly lower than the surface upon which the carapace rested. To initiate the procedure, curved hemostatic forceps (specifically type Rochester-Pean) were introduced through the mouth, ensuring their length was sufficient to reach the distal third of the cervical esophagus, and inserted into the esophagus. The forceps were then firmly pressed against the right lateral wall of the esophagus to identify the protuberance (bump) on the skin surface of the neck (Figure 1).

Before performing the incision, the surgical area was meticulously cleansed using alcohol-free disinfectants, such as povidone-iodine, to ensure aseptic conditions. To minimize discomfort associated with the incision, an infusion of Lidocaine (2%) was administered, diluted with sterile water at 1:1 ratio. This was done to create a local nerve circle block, up to 5 mg/kg total dose [20].

A longitudinal incision measuring approximately 2.5–3 cm in length was carefully made in the neck’s skin, precisely at the point where the forceps caused the protuberance. The pressure exerted by the forceps against the esophagus’s lateral wall served to shift the blood vessels of the neck dorsally, away from the incision site. The subcutaneous tissue was dissected, then the constrictor muscle of the neck was incised, revealing the walls of the esophagus, which were then secured with two or three stay sutures of large caliber (monofilament or coated braided polyester 2 USP) (Figure 2). 

The stay sutures were fixed with two hemostatic forceps; the hemostatic forceps previously inserted through the mouth were carefully removed. Subsequently, the esophagus was gently pulled upward and outward using the stay sutures to apply moderate traction. Following this, a full thickness incision was made in the esophagus wall, just large enough to allow the passage of the esophagogastric tube. After gel lubrication of the tip, the EG tube was carefully inserted through the incision and gently advanced into the esophageal lumen. During this step, potential critical situations (a tube that bends or risks damaging the esophagus) were prevented by applying cranial traction to the right on the esophagus with the use of stay sutures. This helps extend the curved section of the intracelomic esophageal tract slightly forward, thus facilitating the tube’s passage. The use of a rigid nylon stylet inside the tube can be particularly helpful during this process, as it prevents the tip from folding back on itself when it reaches the terminal part of the esophagus. To counter the force of the esophagogastric sphincter, which is particularly tight in sea turtles, it was often necessary to gently maneuver the tube back and forth multiple times, exerting moderate pressure on the esophagogastric sphincter. Pressurized saline in the esophagus through a 60 mL catheter tipped syringe aided the tube insertion, which in this way progressed faster through the esophagogastric sphincter as the saline poured rapidly into the stomach. 

To confirm the correct placement of the catheter, additional physiological solution could be introduced into the tube to ensure that the liquid did not reach the oral cavity. In situations where uncertainties persisted, an intraoperative XR evaluation was conducted. 

To secure the EG tube to the esophagus wall, it was either included in a purse string suture or fixed with two or three interrupted horizontal mattress suture patterns using non-absorbable monofilament material, ranging from USP 2.0 to 1, depending on the size of the animal (Figure 3).

The incision in the skin was then closed around the EG tube, and the tube itself was secured with a Roman purse string suture made of the same material.

Upon completing the surgical procedure, a radiographic check was conducted to confirm the correct positioning of the EG tube (Figure 4).

The outer section of the EG tube was folded back dorsally and secured to the carapace using a self-adhesive elastic bandage that encircled the animal’s body. The end of the tube was sealed with a catheter plug, a Christmas tree adapter, or by placing a syringe barrel inside. Through the extended external EG tube, the animal was fed directly in the water, thereby reducing any additional stress. 

After each feeding session, the EG tube was carefully rinsed with physiological solution to prevent the accumulation of food residues and the potential onset of secondary infections.

Once the sea turtles began feeding independently, we observed a waiting period of 4–7 days before proceeding with the removal of the EG tube.

The removal process and the closure of the esophageal stoma occurs as reported in the literature; any scarring resulting from the tube’s presence and any adhesions formed between the skin and the esophageal wall were scrupulously addressed and removed [10].

### 2.2. Statistical Analysis 

We organized the data using Microsoft Excel and assessed its distribution using the Shapiro test. For data exhibiting a normal distribution, we proceeded with the *T*-test. Conversely, in cases where data did not conform to a normal distribution, we utilized the Kruskal–Wallis test to examine potential group differences.

For all other analyses, we employed the Fisher test and the Kruskal–Wallis test if statistically significant differences were observed. In such cases, we conducted the Dunn test to further investigate the distinctions between groups.

Categorical data were subjected to analysis using contingency tables, and we utilized either chi-squared tests or Fisher’s exact tests for comparisons. Statistical significance was determined at a significance level of *p* < 0.05.

## 3. Results

Regarding the size of the loggerheads that underwent surgery for the EG tube placement, we recorded morphometric measurements including the CCL, ranging from 21 to 83.7 cm (median = 53.0), the CCW, which ranged from 20 to 75.8 cm (median = 49.2), and the weight, ranging from 1.0 to 59.1 kg (median = 16.8) (Table 1). 

The measurements of CCL, CCW, and weight in both facilities exhibited a normal distribution, although the Shapiro test displayed varying degree of variance between the two facilities. In order to evaluate the significance of these variance differences, we applied the Kruskal–Wallis test, which confirmed that the variances were not statistically significant.

In total, 60 cases (39.7%) involved sea turtles affected by both hooks and lines, while 31 turtles (20.5%) had only lines crossing the digestive tract. These particular lesions were more frequently observed at LSTR. Additionally, 34 patients were found to be in very poor condition, in some cases related to entanglement issues (22.5%); exclusively at STC, 26 loggerheads (17.2%) exhibited skull and carapace fractures.

All patients scored between 1 and 3 on the BCS scale; specifically, 55 turtles were classified with a score of 1, 71 turtles with a score of 2, and 25 patients with a score of 3.

The Fisher test confirmed substantial differences in the number of turtles designated for the EG tube procedure across the BCS categories, both in the comprehensive dataset and between the two distinct facilities (Figure 2).

At STC, approximately 20.8% of turtles had a score of 1, 50% scored 2, and 29.2% scored 3. At LSTR, 50.6% were categorized with a score of 1, 44.3% with a score of 2, and 5.1% with a score of 3.

The duration of the EG tube placement for each animal was documented separately for animals that did not survive their original conditions (ranging from 1 to 97 days) and patients that successfully recovered (ranging from 10 to 150 days), followed by a global assessment (Figure 3).

To evaluate the duration of EG tube placement, we opted to include data only from turtles that had recovered (Figure 4 and Table 2).

Because the duration of tube placement did not show a normal distribution in the two facilities, with a prolonged duration at STC (ranging from 17 to 150 days, median = 57.4) compared to LSTR (ranging from 10 to 133 days, median = 25.7), we used the Fisher and chi-squared tests, which confirmed the significant difference in the duration time of EG tube permanence for patients undergoing EG tube procedure in the two facilities (Figure 5).

In case of LSTR, 78.7% of patients required the EG tube for a duration of around 1 month, while in the case of STC, 77.6% needed to keep the EG tube in place for 2 months or longer.

We assessed the significance of the variation in EG tube duration concerning the type of lesions among animals that achieved complete recovery. Specifically, the median duration for turtles with skull and carapace fractures was 69.9 and 86.3 days, respectively. Debilitated turtles had a median duration of 48.2 days, while turtles affected by severe debilitation due to entanglement displayed a median duration of 34.7 days. Turtles with hooks and lines exhibited a median duration of 32.2 days and patients with only lines present in their intestine had a median duration of 25.7 days.

As expected, the majority of fatalities were associated with a BCS of 1 (60.1%), followed by a BCS of 2 (34.5%), and a BCS of 3 (5,4%).

Mortality was observed in 36.4% of the animals, resulting in the loss of four individuals out of 26 with skull and carapace fractures, five patients out of 20 in severely debilitated condition, 11 animals out of 31 related to lines, 28 patients out of 60 due to hooks and lines, and seven turtles out of 14 due to entanglement issues (Figure 6).

### EG-Tube Characteristics and Complications

Through the EG tube, patients received daily feedings consisting of fish smoothies or fish farm meals, and any prescribed therapies tailored to their specific needs were administered (antibiotics, vitamins, minerals, and more).

The esophageal tube remained in situ until the turtle exhibited an interest in food and was capable of spontaneous feeding. The presence of the tube did not hinder animals’ ability to swallow normally. For 63.6%, they transitioned to independent feeding over a span of several weeks; the tubes remained in place for an average duration from approximately 10 to 150 days (mean 57.4 days at STC and 25.7 at LSTR).

The EG tube, despite its relatively inert material, led to moderate local tissue reactions and edema upon contact in six animals. Complications related to the insertion of EG tubes were recorded in 15 sea turtles, constituting 9.9% of cases. These complications typically included some challenges during the insertion of the EG tube into the stomach, as well as its maintenance throughout the recovery period.

The complications included some challenges related to the insertion of the tube, primarily stemming from the strong resistance of the esophago-gastric sphincter. Post-surgery X-rays revealed in seven patients misaligned tube positioning due to folding during insertion, and in two cases, the tube failed to reach the stomach: for those reasons the tube insertion procedure was repeated. In case of accidental tube removal or displacement, frequently triggered by the animals themselves we conducted a brief sedation to reinsert the tube, followed by a purse string suture. We encountered six cases of tube blockage caused by food. For four of these cases, simply rinsing the tube proved effective, but in two cases tube replacement was necessary.

No sea turtle experienced more than one recorded complication. Additionally, there was no statistically significant difference observed in the occurrence of complications across the years of data collection (*p* = 0.543).

No stoma site infections were recorded.

## 4. Discussion

The current study investigates the extensive utilization of EG tubes in sea turtle rehabilitation, conducting a comprehensive analysis on repercussions, potential predisposing factors for complications, and the outcomes associated with maintaining these tubes in situ. The patient age distribution revealed a prevalence of subadult sea turtles at LSTR, which is situated in the central Mediterranean region, while STC, located to the south of the Adriatic Sea and serving as a hub for animals from various rescue centers, predominantly received adult turtles. This appears to align with findings in the literature, which have emphasized the significance of certain regions during different seasons. For instance, the Adriatic Sea has been identified as a crucial area for adult sea turtles [21], while the waters between Sicily and Tunisia are known to be highly frequented [22]. This area borders one of the Mediterranean’s most frequented regions, the Tunisian platform, which is particularly favored by juveniles and subadults [23]. When considering the lesions that afflicted the sea turtles requiring EG tube placement, most cases exhibited a combination of a hook with a line (39.7%) or solely a line present in their digestive tract (20.3%).

Notably, at STC, 14.6% of patients displayed skull fractures, which are more intricate injuries to manage. This factor may explain the higher incidence of cases with a BCS score of 2 for which EG tube placement was deemed necessary at STC, in contrast to the situation at LSTR where the majority of patients had a score of 1. The type of lesion had a statistically significant impact on the duration of EG tube placement. Specifically, in LSTR 78.7% of patients required the EG tube for a duration of approximately 1 month. In contrast, in the case of STC, 77.6% of patients needed to maintain the EG tube in place for 2 months or longer. This difference could possibly be attributed to the presence of specific injuries, such as skull and/or carapace fractures, which are more complex to manage and require a longer recovery period. At STC 20.8% of turtles displayed a BCS of 1, 50% a score of 2, and 29.2% a score of 3, which might be associated with the skull fractures. At LSTR, 50.6% were categorized with a score of 1, 44.3% with a score of 2, and 5.1% with a score of 3. This distribution could be attributed to the logistical complexity involved in securing a surgeon for the procedure, which was limited to only the most critical cases, most of them found floating: probably related to Lampedusa’s extremely isolated geographical location, sea turtles found adrift and in distress often exhibit severe general health conditions. No patients experienced mortality during the EG tube placement procedure, nor did they suffer injuries related to the introduction of the tube. However, 36.4% of sea turtles did not survive in the post-operative period due to their already critical condition upon arrival.

The primary factors contributing to these fatalities were specifically associated with the initial condition of the turtles, which often exhibited severely compromised general condition and lower body condition scores. Furthermore, specific disease processes, like intestinal fishing line entanglement, showed a less favorable prognosis, attributed to the damage to the intestinal and their compromised capacity to absorb food effectively. When reviewing the animal fatalities that occurred following the EG tube placement procedure, as expected, it was evident that patients with a score of 1 constituted a significantly higher proportion of cases (58%) when compared to animals with a score of 2. This discrepancy can be attributed to the fact that the latter group possessed a less compromised physical condition and greater energy reserves, enabling them to better endure both the surgical procedure and the subsequent recovery period. The incidence of fatalities was higher among turtles affected by entanglement issues (53.8%) and the consequences of hooks and lines (46.7%). Interestingly, animals with skull or carapace fractures, despite requiring the EG tube for an extended period, displayed more favorable outcomes, with mortality rates of 13.4% and 23.8%, respectively. This observation may be attributed to their generally better condition, as in many cases the EG tube was employed to assist with debilitated animals experiencing difficulties in self-feeding. The smaller caliber of the EG tube, suitable for small turtles, may result in blockages due to food. This highlights the importance of paying special attention when preparing food like smoothies for administration, ensuring careful consideration of the size and texture of the mixture. Furthermore, it is essential to consider that the tube gauge plays a significant role in the risk of obstruction, underscoring the importance of accurately matching the tube size to the sea turtle’s dimensions. We experienced how EG tubes may play a vital role in providing sea turtle nutritional needs, despite the possibility of associated potential complications.

They are relatively easy to place, with minimal associated risks, and their correct positioning can be confirmed through visual and radiographical examination before the animal regains consciousness from anesthesia. While this study did not document any sea turtle fatalities or significant complications resulting from tube placement, it is crucial to emphasize the importance of ensuring proper procedures to prevent complex outcomes. Generally, the tubes were well tolerated by the animals and remained in place for an average duration of 1 month, with a maximum duration of 150 days (Figure 5).

## 5. Conclusions

This study outlines the utilization of a EG tube procedure in sea turtles as a viable alternative to force-feeding.

It is regarded as a more respectful and suitable approach for the marine turtles, significantly streamlining the daily feeding process. This method permits the direct administration of food into the water of the turtle’s tank, minimizing the stress associated with handling, and preserving operators from the risk of possible injuries during the operations of restraint. Furthermore, essential drugs such as antibiotics, minerals, and vitamins, as well as vital fluids necessary for their clinical management, can be easily administered through this route. Moreover, although not directly observed in our own experience, the use of the EG tube procedure can be pivotal in delivering essential nutritional support to sea turtles during the healing process of mouth-related injuries. Especially for critically ill marine turtles suffering from severe skull or carapace injuries, entanglement of flippers [24,25,26], as well as those with hooks and lines obstructing their digestive tracts [3], the utilization of EG tubes presents numerous advantages over alternative methods of enteral nutrition.

Our experience suggests that EG tubes are generally well tolerated, with no strict limitations on their duration of placement. While the EG tube may not provide a miraculous solution for turtles exhibiting severe emaciation or the inability to absorb nutrients due to their previous compromised health condition, it has proven to be a valuable lifeline for a statistically significant number of patients characterized by extreme fragility. For these individuals, the EG tube has offered the opportunity to absorb essential nutrients and gradually accumulate the energy needed for recovery. Based on our experience, the EG tube has emerged as a pivotal factor for animals in critical condition, making a significant and invaluable difference in their journey toward recuperation from dire situations.

## Data Availability

The data presented in this study are available on request from the corresponding author.

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
