# Peer review of "Esophago-Gastric Tube (EG Tube) in Debilitated Sea Turtle Rehabilitation: Insight in 195 Loggerheads Caretta caretta, Southern Italy"

_vetsci, 2024, doi:10.3390/vetsci11010035_

Round 1

Reviewer 1 Report

Comments and Suggestions for Authors

The information contained in this manuscript is of potentially great value to rehabilitators and veterinarians working with sea turtles, and the description of the surgical procedure is clear and helpful; however, there are some major edits required to be appropriate for publication and useful to readers/clinicians.

It is interesting, clinically, that the authors report that this surgical procedure was typically performed very close to admission for these cases, and with a relatively short duration until the tube was removed, which indicates that a good proportion of these animals started eating on their own within 1-2 weeks of admission anyway. Is undergoing immediate anesthesia and surgery really necessary for animals that never had a chance to actually demonstrate anorexia? 

Many of the graphs and tables are lacking sufficient labels in the axes and legends to be able to understand the information that is being presented.

Most critically, there is little to no insight or investigation into whether the placement of EG tubes had any actual impact on the outcome of the clinical cases. It would be more useful to present additional information on these cases - how were hooks removed? How many turtles underwent other gastrointestinal surgeries, and how did that impact survival?  Is there another cohort of turtles that did NOT receive EG tubes that could be retrospectively compared to these animals to determine if the tube and nutritional support had any impact on overall outcome? 

I would also like to see more details on the complications experienced, especially in terms of maintenance and healing of the tube insertion site and stoma, as the process by which sea turtles heal does not lend itself to this procedure as easily as terrestrial tortoises, and I suspect this is not as benign a process as is presently reported.

Comments on the Quality of English Language

There are significant challenges in following the manuscript, many related to the quality of the English language used, and some could be corrected with careful editing. A few examples are the use of the term 'esophagostomy tube' throughout the manuscript, which typically refers to a tube that is placed with termination in the esophagus, not the stomach, and would result in similar high risk of regurgitation as intermittent tube feeding. What this paper describes is indeed an esophago-gastric tube, which is named in the title, but not elsewhere throughout.  There's also some severe inaccuracies in the description of 'force feeding', which in most contexts would be considered the delivery of whole food items into the esophagus, and in many US facilities is the preferred choice for delivering supplemental nutrition to anorectic sea turtles, very few perform tube feeding as it is described in this paper anymore, except in the most extreme cases.  

Author Response

The information contained in this manuscript is of potentially great value to rehabilitators and veterinarians working with sea turtles, and the description of the surgical procedure is clear and helpful; however, there are some major edits required to be appropriate for publication and useful to readers/clinicians.

  1. It is interesting, clinically, that the authors report that this surgical procedure was typically performed very close to admission for these cases, and with a relatively short duration until the tube was removed, which indicates that a good proportion of these animals started eating on their own within 1-2 weeks of admission anyway. Is undergoing immediate anesthesia and surgery really necessary for animals that never had a chance to actually demonstrate anorexia? 

R: In this paper, our primary focus is on the treatment of sea turtles facing severe anorexia, often resulting from a variety of causes, as evident from their low BCS, which were consistently below 3.

The majority of these animals required surgical intervention frequently related to issues with their esophagus and digestive tract. For these kind of patients, a consistent and well-balanced nutritional support may play a pivotal role, however, the traditional method of force-feeding was deemed unsuitable, especially considering the potential risks associated with inserting a tube into a sutured esophagus, which could lead to dangerous complications. The insertion of the EG tube is performed before the conclusion of each specific surgery required by the animal’s condition.

Historically in our experience prior to 2008, a significant number of these cases resulted in unfortunate fatalities due to the turtles' inability to feed independently. In response to these challenges, we explored alternative approaches, firmly believing that waiting for these turtles to regain independent feeding capabilities was ethically questionable, given their critical condition. In our vision this new approach represents a substantial step forward in the ethical care of these vulnerable creatures.

In the majority of cases, the placement of the esophagostomic tube extended beyond two weeks. The instances of shorter tube permanence were limited to turtles that exhibited swift recoveries, largely owing to effective nutritional support.

  1. Many of the graphs and tables are lacking sufficient labels in the axes and legends to be able to understand the information that is being presented.

R: We appreciate your feedback, and we have taken your advice into consideration. providing more detailed specifications for the content of the graphs and tables. Thank you for your input.

  1. Most critically, there is little to no insight or investigation into whether the placement of EG tubes had any actual impact on the outcome of the clinical cases. It would be more useful to present additional information on these cases - how were hooks removed? How many turtles underwent other gastrointestinal surgeries, and how did that impact survival?  Is there another cohort of turtles that did NOT receive EG tubes that could be retrospectively compared to these animals to determine if the tube and nutritional support had any impact on overall outcome? I would also like to see more details on the complications experienced, especially in terms of maintenance and healing of the tube insertion site and stoma, as the process by which sea turtles heal does not lend itself to this procedure as easily as terrestrial tortoises, and I suspect this is not as benign a process as is presently reported.

R: Due to the emaciated condition of the patients, we consider not ethically viable to establish a control group and this is confirmed by our experience prior 2008: animals with extremely low BCS suffered a high mortality rate due to severe emaciation and inadeguate energy and nutritional support.

The majority of the animals underwent surgeries often related to their digestive tract, and the surgical thecniques and procedures have been extensively documented in multiple papers of the same authors. In this research the primary objective is to emphatize the significance of alternative thecniques for supporting sea turtles in.critical conditions.

Complications we encountered were primarily associated with blockages or the self-detachment of the tube, occasionally due to suture failure, all of which were described.

In all cases, the stoma healed without any complications, even if with varying healing times.

  1. There are significant challenges in following the manuscript, many related to the quality of the English language used, and some could be corrected with careful editing. A few examples are the use of the term 'esophagostomy tube' throughout the manuscript, which typically refers to a tube that is placed with termination in the esophagus, not the stomach, and would result in similar high risk of regurgitation as intermittent tube feeding. What this paper describes is indeed an esophago-gastric tube, which is named in the title, but not elsewhere throughout.  There's also some severe inaccuracies in the description of 'force feeding', which in most contexts would be considered the delivery of whole food items into the esophagus, and in many US facilities is the preferred choice for delivering supplemental nutrition to anorectic sea turtles, very few perform tube feeding as it is described in this paper anymore, except in the most extreme cases.  

R: We apologize for our writing in a non-native language: to enhance the quality of our paper, we have sought the assistance of two English motherlanguage colleagues who are experts in veterinary medicine and biology. They will help refine and improve the text.

In regard to the evaluation of force feeding, we firmly believe that sea turtles encounter diverse emergencies worldwide, and the effectiveness of various techniques can vary depending on the circumstances. In the Mediterranean region, the majority of sea turtles are brought to rescue centers due to injuries caused by bycatch, particularly in longline fishing operations. This often necessitates esophageal surgery, which makes it perilous to introduce a tube into a sutured organ. Moreover, the sea turtles in these cases are typically in critical condition, and in our opinion subjecting them to additional stress could lead to fatal consequences.

Reviewer 2 Report

Comments and Suggestions for Authors

The study describes use of esophagostomy tubes to provide nutritional support and medicine to 195 injured Caretta caretta at two treatment facilities in Italy. The authors provide detailed techniques for surgical placement, describe duration of use as well as complications, and discuss outcomes in relation to categories of injuries and body condition scores, with a comparison between the two sites.

The study provides important information about a technique that can improve treatment outcomes for injured sea turtles. The methods are sound, the results are interesting, and the interpretations are appropriate. Most of my comments and suggestions are relatively minor and are intended to provide clarity.

Comments and suggestions:

L 29: Insert “also” before “involves.”

L34: Suggest “while” rather than “but.”

L47: Phrasing is awkward.

L49: Insert “such” before “as.”

L52: Suggest deleting “even for the” and “the.”

L54: Suggest moving “and energetically balanced” before “diet.”

L67: Suggest “harsh” rather than “rude.”

L86-87: Phrasing is awkward.

Graph 1: Suggest instead: “Number of sea turtles that received EG tube placement” and delete information in parentheses, since it’s redundant with information in graph. Also suggest adding: “Values in bars show overall percentage for each category.” Two significant digits after the decimal seems unnecessary; suggest rounding to tenths. Lastly, the order of categories left to right seems random; perhaps list these alphabetically?

L 152: Suggest deleting “which can be.”

Graph 4: Suggest simply “Body Condition Scores…”

Graph 8: Define values listed above the bars. Also, why are these values shown above bars when other graphs show values within bars, and how was the left to right order of categories determined?

Discussion: Condensing this section into one large paragraph makes it difficult for the reader to assess the information. Suggest separate paragraphs at lines 431, 450, 475.

Conclusions: Much of the information in lines 489-499 seems redundant with information provided later in this section. Consider beginning Conclusions at line 500 (“This study outlines…”) and integrate any essential information from lines 489-499 into this section later. Suggest separate paragraphs at lines 500 and 513.

Comments on the Quality of English Language

Minor editing required.

Author Response

 Thank you, we very appreciate the suggestions and we have changed and fixed the errors. We implemented and revised the discussions and conclusions

Round 2

Reviewer 1 Report

Comments and Suggestions for Authors

There has definitely been some improvement in this manuscript with the current edits, though some challenges remain. I appreciate the revision of the term esophagostomy throughout the manuscript, and I appreciate the insight provided in the letter response that this procedure was most often performed as adjunctive to OTHER surgeries, not entirely on it's own, and I think this manuscript overall would be much better served if that were made much more clear. 

There are still a couple of large issues that extend throughout the whole manuscript that serve to make the information unclear:

The categories of conditions suffered by the animals are very unclear - in lines 123-124, it seems to suggest that the category referred to in all your tables as 'line' might be entanglement in the line and flippers, but much later in the manuscript it seems to be that that group indicates only line in the GI tract (without hooks).  And it's possible the 34 turtles mentioned in lines 123-124 are a combination of the 'debilitated by entanglement' and the 'emaciated' group? Please take the time early on in the writing to clarify exactly what these groups in all of the tables actually are. This also might be made easier to understand by moving the 'line' group next to the 'hook and line' group throughout, as they are similar conditions, and often referred to as closely related later in the manuscript. 

The second real issue is that there is still no description regarding how many of these turtles actually went through additional surgical procedures at the same time as placement of the EG tube, which I would consider a critical piece of information. The language of lines 157-170 could all be modified to make it more clear that this was an adjunctive procedure in those cases, as the way it is written right now, it's confusing about whether they were separate procedures that occurred prior or after other surgeries. 

You note in your response letter that most turtles had tubes in for a longer time than I implicated, but with a median tube placement time of 20 days at LSTR, and a 4-7 day waiting period after the animal began eating on its own, that's a median anorexia period of about 14-16 days - not really very long in 'turtle time'.  HOWEVER, I can completely understand and respect your clinical opinion that for turtles that have undergone esophageal or GI tract surgery, this is likely a safer and easier way to deliver post-operative nutrition, so I really think that's a stronger argument that you should be making.  But nowhere in the manuscript does it ever indicate how many of these cases actually underwent such surgeries of the GI tract - was it all of them with hooks and/or lines? (e.g. in lines 401-403 would be a good place to make this clear). 

A few other specific suggestions:

Line 150: what exactly were the findings that determined a patient to be suitable for this procedure? If it was simply poor body condition, regardless of cause, then this statement has no place.

Line 217: what are the 'potential critical situations' being avoided here?

Lines 256-258: I assume turtles were anesthetized again for tube removal? A small amount of expansion on the removal process would be good here. Esophagus and skin closed separately, I assume? 

Line 425: in complications, what was done about the tissue reactions, if anything? What was done when turtles displaced the tubes themselves - were those replaced in another procedure?

Lines 469-472: is this sentence referring to the conditions of the turtles, or the availability of surgeons to perform this procedure? It's not clear.

Line 492: discusses the potential of tube blockages, but this was not mentioned elsewhere in the manuscript - did this ever occur? And if so, how was it addressed by the authors?

Graphs/figures comments:

Graph 2 is not really necessary and doesn't convey any information that is not readily available in the text.

Graph 4: I still have no idea what information this is meant to convey -the text is too small and the individual graphs are not labeled

Graph 7: is this representing mortality or success numbers? It doesn't specify.

Comments on the Quality of English Language

It is clear that your English as a first language colleagues assisted greatly in revising the beginning of the manuscript, but these efforts seemed to fall off as the manuscript proceeded. A few specific instances that could use correction:

I still question the use of the term 'force feeding' over 'tube feeding', as they are two very different procedures applicable to different situations (and I fully agree that tube feeding as you describe it poses substantial risks that are often not worth it).

Line 166: remove 'thanks to' , and replace with something like 'using'.

Line 186: change 'were' to 'and'

Lines 338...: multiple instances of 'the' in front of numerical values where it does not belong

Author Response

“Esophago-gastric tube (EG tube) in debilitated sea turtle rehabilitation: insight in 195 loggerheads Caretta caretta, Southern Italy”

Dear editor,

On the behalf of the Authors, we would like to express our gratitude for the comments and suggestions provided, as they have proven to be valuable in enhancing the quality of our manuscript. While we do have some partial agreements with the reviewers’ comments, we have taken their suggestions into careful consideration and made the necessary amendments to the revised version of the manuscript, as outlined in our revisioned manuscript.

We sincerely believe that the manuscript is now well-prepared for publication.

Once again, thank you for your editorial guidance.

Sincerely

Prof. Stefano Ciccarelli

REV 1

There has definitely been some improvement in this manuscript with the current edits, though some challenges remain. I appreciate the revision of the term esophagostomy throughout the manuscript, and I appreciate the insight provided in the letter response that this procedure was most often performed as adjunctive to OTHER surgeries, not entirely on it's own, and I think this manuscript overall would be much better served if that were made much more clear. 

There are still a couple of large issues that extend throughout the whole manuscript that serve to make the information unclear:

The categories of conditions suffered by the animals are very unclear - in lines 123-124, it seems to suggest that the category referred to in all your tables as 'line' might be entanglement in the line and flippers, but much later in the manuscript it seems to be that that group indicates only line in the GI tract (without hooks).  And it's possible the 34 turtles mentioned in lines 123-124 are a combination of the 'debilitated by entanglement' and the 'emaciated' group? Please take the time early on in the writing to clarify exactly what these groups in all of the tables actually are. This also might be made easier to understand by moving the 'line' group next to the 'hook and line' group throughout, as they are similar conditions, and often referred to as closely related later in the manuscript. 

R1: Thank you for your valuable suggestions; we appreciate them. We have addressed the issues and misunderstandings by making the necessary changes and improvements to this section.

The second real issue is that there is still no description regarding how many of these turtles actually went through additional surgical procedures at the same time as placement of the EG tube, which I would consider a critical piece of information. The language of lines 157-170 could all be modified to make it more clear that this was an adjunctive procedure in those cases, as the way it is written right now, it's confusing about whether they were separate procedures that occurred prior or after other surgeries. 

You note in your response letter that most turtles had tubes in for a longer time than I implicated, but with a median tube placement time of 20 days at LSTR, and a 4-7 day waiting period after the animal began eating on its own, that's a median anorexia period of about 14-16 days - not really very long in 'turtle time'.  HOWEVER, I can completely understand and respect your clinical opinion that for turtles that have undergone esophageal or GI tract surgery, this is likely a safer and easier way to deliver post-operative nutrition, so I really think that's a stronger argument that you should be making.  But nowhere in the manuscript does it ever indicate how many of these cases actually underwent such surgeries of the GI tract - was it all of them with hooks and/or lines? (e.g. in lines 401-403 would be a good place to make this clear). 

R1: As specified in the text, all turtles underwent surgery for their condition, and the placement of the EG tube was performed as an adjunctive procedure during the scheduled surgeries. We have enhanced the description of this process in line with your suggestions.

Additionally, considering the remote location of Lampedusa, which is only accessible by a small aircraft for a significant portion of the year, surgeons were only able to travel under favorable conditions.

We have included all the information as requested.

A few other specific suggestions:

Line 150: what exactly were the findings that determined a patient to be suitable for this procedure? If it was simply poor body condition, regardless of cause, then this statement has no place.

Line 217: what are the 'potential critical situations' being avoided here?

Lines 256-258: I assume turtles were anesthetized again for tube removal? A small amount of expansion on the removal process would be good here. Esophagus and skin closed separately, I assume? 

Line 425: in complications, what was done about the tissue reactions, if anything? What was done when turtles displaced the tubes themselves - were those replaced in another procedure?

Lines 469-472: is this sentence referring to the conditions of the turtles, or the availability of surgeons to perform this procedure? It's not clear.

Line 492: discusses the potential of tube blockages, but this was not mentioned elsewhere in the manuscript - did this ever occur? And if so, how was it addressed by the authors?

Graphs/figures comments:

Graph 2 is not really necessary and doesn't convey any information that is not readily available in the text.

Graph 4: I still have no idea what information this is meant to convey -the text is too small and the individual graphs are not labeled

Graph 7: is this representing mortality or success numbers? It doesn't specify.

R1: We appreciated, incorporated and revised all the suggestions that didn’t change the aim of our manuscript.

Comments on the Quality of English Language

It is clear that your English as a first language colleagues assisted greatly in revising the beginning of the manuscript, but these efforts seemed to fall off as the manuscript proceeded. A few specific instances that could use correction:

I still question the use of the term 'force feeding' over 'tube feeding', as they are two very different procedures applicable to different situations (and I fully agree that tube feeding as you describe it poses substantial risks that are often not worth it).

R1: It is a common practice in rescue centers to use assisted feeding, which involves the daily administration of nutrition through a soft tube passed from the mouth to the stomach. While this procedure is relatively straightforward for most reptile species, it can be quite challenging in chelonians, especially large sea turtles. This is because access to the mouth and esophagus can become obstructed if the animal retracts its head. Furthermore, this practice can be a significant source of stress for wild animals, and in the case of sea turtles, it can be messy due to their particularly narrow gastroesophageal sphincter.

We understand the reviewer's concern, but we have chosen to retain the terminology used in "Sea Turtle Health and Rehabilitation" by Manire C.A. et al. This terminology is commonly employed by individuals involved in rehabilitation efforts.

Line 166: remove 'thanks to' , and replace with something like 'using'.

Line 186: change 'were' to 'and'

Lines 338...: multiple instances of 'the' in front of numerical values where it does not belong

R1: Thank you, we have modified and reviewed our paper with all your suggestions.
